# Barriers to Postpartum Glucose Intolerance Screening in an Italian Population

**DOI:** 10.3390/ijerph15122853

**Published:** 2018-12-14

**Authors:** Paola Quaresima, Federica Visconti, Eusebio Chiefari, Luigi Puccio, Daniela P. Foti, Roberta Venturella, Raffaella Vero, Antonio Brunetti, Costantino Di Carlo

**Affiliations:** 1Unit of Obstetrics and Gynecology, Department of Medical and Surgical Sciences, University “Magna Græcia’’ of Catanzaro, Viale Europa, 88100 Catanzaro, Italy; dr.paolaquaresima@gmail.com (P.Q.); fed.visconti@gmail.com (F.V.); rovefa@libero.it (R.V.); 2Department of Health Sciences, University “Magna Græcia” of Catanzaro, Viale Europa, 88100 Catanzaro, Italy; echiefari@libero.it (E.C.); foti@unicz.it (D.P.F.) brunetti@unicz.it (A.B.); 3Complex Operative Structure Endocrinology-Diabetology, Hospital Pugliese-Ciaccio, 88100 Catanzaro, Italy; puccio55@libero.it (L.P.); rafvero@libero.it (R.V.)

**Keywords:** gestational diabetes mellitus, ppOGTT, type 2 diabetes mellitus

## Abstract

*Background*: Gestational diabetes mellitus (GDM) is a strong risk factor for type 2 diabetes mellitus (T2D) and the postpartum period is crucial for early treatment in at-risk women. However, despite recommendations, only a fraction of women undergo a postpartum screening for glucose intolerance (ppOGTT). The present study aims to verify the reason(s) for poor adherence in our population. *Research design and methods*: This retrospective study includes 451 women in which GDM was diagnosed between 2015–2016. During 2017, we verified by phone interview how many women underwent ppOGTT at 6–12 weeks postpartum, as recommended by the Italian guidelines. The non-compliant women were asked about the reason(s) for failing to screen. The non-parametric Mann-Whitney test and the 2-tailed Fisher exact test were used to compare continuous and categorical features, respectively, among women performing or non-performing ppOGTT. *Results*: Out of 451 women with GDM diagnosis, we recorded information from 327. Only 97 (29.7%) performed ppOGTT. The remaining 230 women (70.3%) provided the following explanation for *non-compliance*: (1) newborn care (30.4%); (2) misunderstood importance (28.3%); (3) oversight (13.0%); (4) unavailability of test reservation in the nearest centers (10.4%); (5) normal glycemic values at delivery (8.3%); (6) discouragement by primary care physician (5.6%). *Conclusions*: In our population, most women with recent GDM failed to perform ppOGTT. Our results indicated that the prominent barriers could potentially be overcome.

## 1. Introduction

Type 2 diabetes mellitus (T2D) is the most common metabolic disease. It affects about 9% of the adult population [1], with an increasing global incidence due to the westernized lifestyle [2]. T2D is associated with long-term complications which greatly affect quality of life, reduce life expectancy, and contribute to the huge healthcare costs [3,4]. Therefore, it is mandatory to implement rigorous strategies to prevent T2D, to reduce its risks, and to address diabetic patients towards early and efficacious therapies.

Gestational diabetes mellitus (GDM), the type of glucose intolerance that develops during the second and third trimester of pregnancy [5], is a strong risk factor for the development of T2D later in life. In fact, T2D has a cumulative incidence of 60% at 10 years after a pregnancy, complicated by GDM [6]. Also, the worldwide incidence of GDM has greatly risen, affecting 20% of the pregnant women tested in some populations [7]. Many lines of evidence indicate that the identification of women with glucose intolerance during the postpartum period is of critical importance, as appropriate treatments can prevent or delay the onset of T2D [8,9,10]. Therefore, the most important medical societies all recommend screening for glucose intolerance in the early post-partum period in women with GDM [11]. Nevertheless, no consensus exists about when and how this screening should be performed [11]. In our country, current guidelines recommend that women with GDM have a 2-h 75 g oral glucose tolerance test (OGTT) 6 to 12 weeks after delivery (ppOGTT) [12]. However, despite compelling lines of evidence demonstrating the relevance of postpartum screening among women with previous GDM and the increased risk for non-adherent women in developing T2DM later in life, the rate of GDM women receiving appropriately-timed postnatal glucose testing is quite low [11]. Several explanations have emerged for the non-attendance of women for ppOGTT after GDM [13,14,15,16,17,18]. They include patient subjective barriers, such as newborn care, travel and socioeconomic difficulties, the lack of interest for healthcare, the lack of family support, the lack of understanding about the T2D risk and test discomfort [19]. They also include concerns regarding the healthcare system, including insufficient communication between gynecologists/diabetologists and primary care physicians, the perception among gynecologists, and primary care physicians that postpartum follow-up of GDM is not important, as well as the lack of universal guidelines [20]. In order to overcome barriers of testing, many interventions have been proposed in recent years [21]. Among them, it should be mentioned that verbal and written antepartum counselling [21,22], continuous postpartum follow-up [12], patient and physician reminders [14,23,24,25,26,27], flexible appointment times and dynamic roles in decision and planning of medical tests [28]. However, none of these strategies have obtained satisfying results [11]. We previously reported that adopting verbal and written antepartum counselling increased the follow-up screening rate from 24.1% to 62.4% in our population [22], where GDM prevalence is relatively high [29]. Unfortunately, this procedure, which has been derived from a specific healthcare project, has been stopped for economic reasons. Thus, we planned to retrospectively verify the adherence rate to ppOGTT and the reasons for non-adherence.

## 2. Materials and Methods

From January 2015 to December 2016, 1413 pregnant women underwent GDM screening at the Hospital “Pugliese-Ciaccio” in Catanzaro, Italy, in compliance with the Italian guidelines [30]. Out of the 1413 women, 451 (31.8%) were diagnosed with GDM. The diagnosis of GDM was made in accordance with the IADPSG cut-off points (fasting ≥ 92 mg/dl, 1h ≥ 180 mg/dl, 2h ≥ 153 mg/dl) [31]. Women with pre-existing type 1 or type 2 diabetes mellitus, as defined by ADA criteria [5], were excluded from the study. Anamnestic information included age, parity, family history of diabetes (first- or second-degree relatives), previous GDM, self-reported pre-pregnancy weight, pre-existing polycystic ovary syndrome (PCOS) (as defined by “The Rotterdam ESHRE/ASRM-sponsored PCOS consensus workshop group” criteria) [32], class of GDM risk (as defined by Italian guidelines) [30], insulin treatment. From all women, the information above were recorded at the time of GDM screening and during the gestation. This cohort has already been included in a previous work [33]. All interviews were conducted by the principal investigators by telephone and then transcribed. The interview was structured with closed-ended questions. For each case, only the main barrier was recorded. All women provided a verbal consent before the interview. The study was approved by the local ethics committee (n° 33, 27 June 2017).

After testing for normality of all continuous variables by the Shapiro-Wilk normality test, the non-parametric Mann-Whitney test was used for comparisons among the women performing or not-performing ppOGTT. The 2-tailed Fisher exact test was employed for comparison of proportions. Continuous variables are expressed as median and interquartile range (IQR), and categorical variables as numbers and percentages. In all analyses, statistical significance was fixed at an alpha level of 0.05. All calculations were performed with SPSS 20.0 software (SPSS Inc, Chicago, IL, USA).

## 3. Results

We were able to contact 327 (72.4%) out of 451 pregnant women with recent GDM. Among them, 230 (73.3%) did not perform ppOGTT, and only 97 (29.7%) were compliant with the Italian guidelines. As shown in Table 1, the return rate was higher in women classified as “high risk” for GDM during pregnancy, in those undergoing early GDM screening, with familial history of T2D in first grade relatives, with previous GDM, with middle/high educational status, and with PCOS. No difference was observed when age, pre-pregnancy BMI, gravidity, and insulin treatment were considered.

During the phone interviews, the women that had not performed ppOGTT were asked for the reasons of their poor compliance, and all their answers were classified into 6 groups (Table 2). As reported in Table 2, the first barrier that prevented postnatal screening was newborn care. In fact, 30.4% of women claimed that the need for continuous baby care was the main reason that had prevented them from ppOGTT testing. The second barrier was due to their poor understanding of the importance of this test (28.3%). Then, 30 (13.0%) cases admitted that they had overseen it; 24 (10.4%) women were unable to obtain a ppOGTT reservation in the nearest centers. In Italy, the public healthcare system requires a reservation to have access to any health service, but the waiting lists are often very long. Of the cohort, 19 (8.3%) women believed that the normalization of glycemic values after the delivery made the ppOGTT unnecessary. Surprisingly, 13 (5.6%) women claimed that they have been discouraged by their primary care physician. When we searched a correlation between the different barriers and the women’s features, we observed that, among the above-mentioned barriers, newborn care was more frequent among women with a GDM requiring insulin treatment compared with insulin-untreated women (31/67 versus 39/163, *p* = 0.0015). No other correlation was observed.

## 4. Discussion

Current Italian guidelines recommend ppOGTT in women with GDM. However, we found that the adherence to this test has decreased since antepartum counseling has been left out [22]. Herein, we investigated the barriers to ppOGTT in Italian women, after the interruption of verbal antepartum counseling [22]. We first observed that 70.3% (230/327) of women with recent GDM did not perform ppOGTT. That is, the rate of women performing ppOGTT has been halved with respect to that observed when counselling was introduced [22], thus confirming the critical importance of this tool in our population. The investigation of obstacles to ppOGTT indicated that a prominent barrier was represented by a competing priority, that is newborn care. This data is consistent with other reports and, interestingly, it is common to any examined population [18,34,35]. In addition, it is worth noting that the priority towards the newborn constitutes a prevailing barrier even in women treated with insulin during pregnancy, a category that generally demonstrates a stronger compliance for ppOGTT.

Similar to other reports, our findings indicated that the lack of patient understanding and awareness of the risks of T2DM and the low motivation for self-care are relevant reasons for non-adherence to ppOGTT [22,36,37,38,39]. In fact, in our study, over 50% of answers (answers 2), 3), and 5)) can be attributed to these reasons. This is consistent with the observations that women with middle/high educational status and/or with a previous clinical condition are predisposed to T2DM susceptibility (e.g., previous GDM, PCOS, high risk for GDM), all factors already stimulating a greater awareness, were more compliant. Furthermore, this study is consistent with previous data, from us and others, indicating a marked improvement in return rate after appropriate counselling or other education tools [11]. The difficulty in obtaining a ppOGTT reservation in the nearest centers is a structural obstacle, very frequent in Southern Italy, that is related to the effectiveness and efficiency of healthcare system organization. However, we believe that even this barrier may be at least in part affected by the lack of awareness and low motivation for self-care. Inadequate communication between obstetrician and primary care physician, the perception that postpartum follow-up of GDM is not a clinical priority, and the lack of agreed protocols/procedures may be responsible for the last barrier [39,40]. Since failure to perform ppOGTT testing may potentially prevent the onset or delay of the diagnosis of T2DM in women who experienced GDM, and since compliance may be overcome by antepartum counselling, efforts should be made to encourage this practice in Southern Italy.

## 5. Conclusions

In our population, most women with recent GDM failed to perform the ppOGTT testing. The lack of patient understanding and awareness of the risks leading to T2DM, the low motivation for self-care and inadequate counselling by primary care physicians are contribute to non-adherence. All these barriers can be potentially overcome. Based on this study, we have promoted a patient’s specific informative consultation close to term pregnancy and provided information protocols on ppOGTT to primary care physicians of our district. Further studies are needed to evaluate the efficacy of these methods for improving patients’ compliance, and to establish whether counselling may be a good practice that could be generalized to other populations.

## Figures and Tables

**Table 1 ijerph-15-02853-t001:** Characteristics of the enrolled women who had, or had not performed ppOGTT.

Maternal Characteristics	ppOGTT	No ppOGTT	*p* Value
	*n* = 97	*n* = 230	
Age, yr	33 (30–36)	34 (30–36)	0.8842
Pregravidic BMI, kg/m^2^	25.9 (22.3–28.3)	25.1 (22.4–28.2)	0.6229
Familial history of type 2 diabetes, *n*	52 (53.6%)	87 (37.8%)	0.0101
Previous GDM, *n*	21 (21.6%)	27 (11.7%)	0.026
Gravidity, *n*	2 (1–2)	2 (1–2)	0.6464
Middle/high educational status, *n*	71 (73.2%)	105 (45.6%)	<0.0001
PCOS, *n*	17 (17.5%)	11 (4.8%)	0.0004
High risk women *, *n*	47 (48.4%)	42 (18.3%)	<0.0001
Early GDM screening *, *n*	58 (59.8%)	31 (13.5%)	<0.0001
Insulin treatment, *n*	38 (39.2%)	67 (29.1%)	0.0917

Data are medians (interquartile range in brackets) for age, pre-gestational BMI, and gravidity or numbers (*n*). P values refer to overall differences across groups as derived from non-parametric Mann Whitney test or Fisher’s exact test, respectively. Middle/high educational status: Secondary school/university degree. * According to the Italian guidelines, high risk women are those with at least one of the following parameters: previous GDM, pre-pregnancy body mass index (BMI) ≥ 30 kg/m^2^, fasting plasma glucose (FPG) at first visit or before pregnancy between 100–125 mg/dL (5.6–6.9 mmol/L). For these women, a 75-g 2 h-OGTT is recommended early in pregnancy (16–18 weeks) [30].

**Table 2 ijerph-15-02853-t002:** Prominent barriers to ppOGTT.

Barrier Reference Number	Barrier	*n* = 230
1	Newborn care	70 (30.4%)
2	Misunderstood the importance of ppOGTT	65 (28.3%)
3	Overseen	30 (13.0%)
4	Unavailability of ppOGTT reservation in the nearest centers	24 (10.4%)
5	Normal glycemic values after delivery	19 (8.3%)
6	Advised against by primary care physician	13 (5.6%)

Data are numbers (*n*).

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
