# Peer review of "Barriers to Postpartum Glucose Intolerance Screening in an Italian Population"

_ijerph, 2018, doi:10.3390/ijerph15122853_

Reviewer 1 Report

In this retrospective study, the authors planned to retrospectively verified the adherence rate to postpartum screening for glucose intolerance (ppOGTT) and the reasons of non-adherence. The authors concluded that patient understanding and awareness of the risk in relation to T2DM, the low motivation for self-care and the inadequate counselling by primary care physician are relevant reasons for non-adherence.

Comments

This is an interesting retrospective study. The manuscript is well-written. The reviewer has only some minor concerns as follows:

1. The footnotes in Table 1 and Table 2 are not clear; some indications are lacking such as symbol “*”, N number, or %.

2. In BMI, kg/m2 (Table 1 and line 108), “2” needs to be superscript.

3. In line 58, reference [13-8] needs to be revised to [13-18].

Author Response

Reply to Reviewer 1:

Thanks for the kind comment

1) As requested, the footnotes in Table 1 and Table 2 have been clarified. The lacking indications have been added.

2).We superscripted  “2” in  BMI, kg/m2 both for Table 1 and line 108 

3) In line In 58, we revised the reference from  [13-8]  to [13-18].

ethical code for our research 

inner Number: 33, 27 June 2017

Reviewer 2 Report

Abstract

Abstract should be accurately reflect the manuscript as a whole, methods section need to be revised and expand in an academic writing. Second, author may wish to add couple of lines of statistical analysis/test used in the study.

Introduction

The introduction provides a good, generalized background of the topic. The authors have introduced the related work clearly.

Methods

The authors did not give information about patient informed consent. Second, Authors may wish to expand or provide details of interview say structured/ non structured, questions were open ended or not. This is not purely retrospective study as patients were identified via data base and then contacted for their feedback/perception and practice regarding ppOGTT. 

Results

Table 1 is quite confusing probably need to be revised as percentages are conflicting for example High risk women*, N 47 (48.4)     42 (18.3)     <0.0001 (Though % I column is standard way to report, However in this table authors may wish to provide % in rows.

Please clearly explain where median and IQR used and when numbers and % were used as all clinician are not good in statistic.

Discussion

The discussion should be of great interest to the readers. However, is not well written hence need to revise. The discussion section offers the “how” and “why” explanations for study findings. It is also a place for authors to consider “glaring” elements of their data and findings.

Author Response

Reply to Reviewer 2:

As requested, abstract methods section has been revised and expanded in an academic writing, moreover some lines about the adopted statistical analysis has been added.

Many thanks for your comment about the introduction.

Methods section has been revised and extended. In particular, the employed statistical analysis has been explained on  page 1, lines 24-27,

additional information about patient informed consent and interviews structure have been added on page 2, lines 87-89.

According to the Results requested reviews Table 1 and Table 2 have been modified, as well as their footnotes. Finally the Discussion has been revised and expanded.

Reviewer 3 Report

The manuscript is generally well written and presented.  The intent is to examine barriers to postpartum OGTT screening at 6-12 weeks. Data are retrospective and interview collected.    

Major 

No major concerns

Minor 

Table 2 is missing a caption

Document should be edited for English to correct minor phrasing and verb tense errors

Author Response

Thank you for our paper appreciation,

As requested we added a caption to table 2 and reviewed English to correct minor phrasing and verb tense errors.